# PGRS Domain of Rv0297 of *Mycobacterium tuberculosis* Functions in A Calcium Dependent Manner

**DOI:** 10.3390/ijms22179390

**Published:** 2021-08-30

**Authors:** Tarina Sharma, Jasdeep Singh, Sonam Grover, Manjunath P., Firdos Firdos, Anwar Alam, Nasreen Z. Ehtesham, Seyed E. Hasnain

**Affiliations:** 1Kusuma School of Biological Sciences, Indian Institute of Technology, Hauz Khas, New Delhi 110016, India; tarina.sharma2@gmail.com (T.S.); firdos184@gmail.com (F.F.); 2ICMR-National Institute of Pathology, Safdarjung Hospital Campus, New Delhi 110029, India; manjusabu114@gmail.com (M.P.); dranwar.iit@gmail.com (A.A.); 3JH-Institute of Molecular Medicine, Jamia Hamdard, Tughlaqabad, New Delhi 110062, India; jasdeep002@gmail.com (J.S.); sonamgbt@gmail.com (S.G.); 4Department of Biochemical Engineering and Biotechnology, Indian Institute of Technology, Hauz Khas, New Delhi 110016, India; 5Dr. Reddy’s Institute of Life Sciences, University of Hyderabad Campus, Prof C.R. Rao Road, Hyderabad 500019, India

**Keywords:** mycobacteria, tuberculosis, calcium, cytokine release, intrinsically disordered proteins, nitric oxide, PE_PGRS, TLR4 signaling, host-pathogen interaction

## Abstract

*Mycobacterium tuberculosis (M.tb)*, the pathogen causing tuberculosis, is a major threat to human health worldwide. Nearly 10% of *M.tb* genome encodes for a unique family of PE/PPE/PGRS proteins present exclusively in the genus Mycobacterium. The functions of most of these proteins are yet unexplored. The PGRS domains of these proteins have been hypothesized to consist of Ca^2+^ binding motifs that help these intrinsically disordered proteins to modulate the host cellular responses. Ca^2+^ is an important secondary messenger that is involved in the pathogenesis of tuberculosis in diverse ways. This study presents the calcium-dependent function of the PGRS domain of Rv0297 (PE_PGRS5) in *M.tb* virulence and pathogenesis. Tandem repeat search revealed the presence of repetitive Ca^2+^ binding motifs in the PGRS domain of the Rv0297 protein (Rv0297PGRS). Molecular Dynamics simulations and fluorescence spectroscopy revealed Ca^2+^ dependent stabilization of the Rv0297PGRS protein. Calcium stabilized Rv0297PGRS enhances the interaction of Rv0297PGRS with surface localized Toll like receptor 4 (TLR4) of macrophages. The Ca^2+^ stabilized binding of Rv0297PGRS with the surface receptor of macrophages enhances its downstream consequences in terms of Nitric Oxide (NO) production and cytokine release. Thus, this study points to hitherto unidentified roles of calcium-modulated PE_PGRS proteins in the virulence of *M.tb*. Understanding the pathogenic potential of Ca^2+^ dependent PE_PGRS proteins can aid in targeting these proteins for therapeutic interventions.

## 1. Introduction

*Mycobacterium tuberculosis*, the pathogen causing tuberculosis (TB), is responsible for nearly 1.6 million deaths, with more than 10 million new cases of TB globally. Drug-resistant strains of *M.tb,* such as Multi drug resistant (MDR) and extensive drug resistant (XDR), have further worsened the present scenario of TB in India, with 24% of total drug-resistant cases globally [1,2,3]. *M.tb*, being an intracellular pathogen, has evolved several virulence characteristics to survive and disseminate in host macrophage [4]. Approximately 10% of the *M.tb* genome codes for PE/PPE/PE_PGRS protein family [5]. Expansion in the PE/PPE/PE_PGRS family of proteins is an exception for the otherwise reductive evolution of the *M.tb* genome, which points towards the pathogenic roles of these proteins during *M.tb* infection [6,7,8,9]. The PE_PGRS proteins possess intrinsically disordered regions that are critical to their activity [10]. The presence of such intrinsically disordered regions in *M.tb* proteome may assist in subverting host immune mechanisms [8,9,10]. Changes in the secondary structures of the PE/PPE/PE_PGRS proteins from unstructured to structured patterns have been hypothesized to play a role in the progression of disease pathogenesis and modulation of host cellular responses [11]. The PE_PGRS proteins generally exhibit tandem repeats of Gly-Gly-Ala or their variants [6,7]. Within the different PE_PGRS proteins, variations in size and number of the repetitive sequences of Gly-Gly-Ala or Gly-Gly-Asn motifs [12] play a critical role in antigenicity [13] and evasion of host immune response [8,12,14,15,16]. PE_PGRS proteins are known to serve multiple functions, including pro- and anti-inflammatory immune responses [13,17,18,19], host cell apoptosis [20,21], bacillary survival [22], granuloma maintenance [23], and phagolysosome formation [24]. Several of PE_PGRS proteins of *M.tb* H_37_Rv and H_37_Ra exhibit genomic and proteomic differences that point to the role of these proteins in virulence and pathogenicity [25]. The well-studied protein PE_PGRS33 is known to initiate *M.tb* entrance into macrophages via its interaction with TLR2 [26]. PE_PGRS33 localizes to mitochondria, induces host cell apoptosis [20,27,28,29,30,31], and evokes pro-inflammatory cytokine response from host macrophages [17,29]. The modulation of microbial growth and anti-inflammatory immune response of macrophages due to PE_PGRS30 in terms of reduced production of IL-12, TNF-α, and IL-6 has been reported [32,33]. The sub-family of PE_PGRS proteins was found to be exclusively present in pathogenic strains of mycobacterium such as *M.tb, M. marinum* and *M. bovis* [12]. Their absence in non-pathogenic strains points to their prospective significance in tuberculosis pathogenesis and virulence.

The development of the disease and emergence of drug resistance is determined by the interplay of host-pathogen interactions [2,34,35], also involving apoptosis, necrosis, and autophagy [36,37,38]. Ca^2+^ is crucial as a secondary messenger and it regulates several physiological processes in host immune cell responses [39]. The intracellular and extracellular Ca^2+^ ions both bind to cytosolic proteins and cell-surface proteins, respectively, and modulate cellular signaling [40]. During the infection process, numerous bacterial proteins have the capability to bind to calcium ions for modulating host cellular responses, resulting in disease pathogenesis [41,42]. The ability of *M.tb* proteins to bind to Ca^2+^ allows it to regulate host cellular processes such as apoptosis, acidification of phagolysosomes, generation of NO intermediates and Reactive Oxygen Species (ROS), and interaction with TLRs [43,44].

PE_PGRS5 of *M.tb* is coded by the Rv0297 gene, and its PGRS domain (Rv0297PGRS) consists of 15 Ca^2+^-binding motifs of GGXGXD/NXUX type. We previously demonstrated that the Rv0297PGRS induces endoplasmic reticulum (ER) stress-mediated apoptosis in a TLR4 dependent manner [21]. The PGRS domain of Rv0297 has also been described in a previous report to modulate macrophage functions and enhance bacillary survival [45]. We hypothesize that the Ca^2+^ dependent binding of Rv0297PGRS with host cell receptors results in initiating cellular signaling that is implicated in disease pathogenesis. In the present study, we show that Ca^2+^ ions stabilize binding of Rv0297PGRS with TLR4 of macrophages. Using MD simulations and biophysical characterization, we have elucidated that Ca^2+^ ions were able to provide moderate stabilization to the overall architecture of protein under thermal denaturing stresses. This Ca^2+^ dependent interaction of Rv0297PGRS with TLR4 on macrophages was also responsible for inducing NO production. Rv0297PGRS was able to evoke a pro-inflammatory response by inducing the increased production of TNF-α and IL-12 cytokines in a calcium-dependent manner. Our results highlight the novel functions of a PE_PGRS protein Rv0297PGRS as a calcium-dependent protein that is involved in pathogenesis.

## 2. Results

### 2.1. PGRS Domain of M.tb Rv0297 Contains Putative Calcium-Binding Motifs

Tandem repeats of GG-X-GX-D/N-XUX type putative calcium-binding motifs (where X is any and U is a large non-polar amino acid), PGRS domains of PE_PGRS family of *M.tb* were analyzed, as described previously [46]. The composition/type and number of such repeats varied among the different PE_PGRS protein sequences (Appendix A).

We observed 15 putative Ca^2+^ binding nonapeptide motifs in the Rv0297PGRS protein sequence (Figure 1A). This was comparatively higher than the previously known Ca^2+^ binding PE_PGRS33 (10 motifs), PE_PGRS20 (10 motifs), and PE_PGRS45 (13 motifs) proteins of *M.tb*. However, multiple sequence alignment showed the conservation of these motifs (Figure 1A). During I-TASSER modeling, these regions were also predicted in the Rv0297PGRS 3D model by COACH and IonCom functional annotations. Although secondary structure annotations of the Rv0297PGRS by I-TASSER, PSIPred, and IUPred predicted its intrinsically disordered nature, the deconvolation of its CD spectra showed ~44% β-content in the protein (Appendix A). Given the limitations of homology modelling, different modelling programs were used for structural determination. However, the given full length Rv0297PGRS model (predicted through i-tasser) held close proximity to CD driven secondary structure estimation, and it was chosen for further analysis. The predicted model showed distinct N-terminal and C-terminal domains that were connected through a short linker (Gly^425^–Gly^430^) (Figure 1B). Surface electrostatics showed various negatively charged regions, distributed in both the N-terminal and C-terminal domains (Figure 1C). 

### 2.2. Ca^2+^ Binding Stabilizes PGRS Domain Architecture

Previous studies on Ca^2+^ binding showed that Ca^2+^ induced the folding and stabilization of various intrinsically disordered proteins [39,41,42]. Hence, we wanted to ascertain the effects of Ca^2+^ ions on the Rv0297PGRS functioning. In concurrence with the predicted intrinsic disordered nature, the simulations of Rv0297PGRS alone showed an increase in the backbone RMSD and gyration radius over the entire simulation period (Figure 2A,B). However, in the presence of Ca^2+^, we observed low RMSD and gyration radius when compared to PGRS alone. To map all possible conformations of Rv0297PGRS that were adopted during both simulations, free energy landscapes (FEL) were constructed by projecting these two principal components. The lowest free energy conformations of Rv0297PGRS alone were characterized by a structure with high RMS deviation from the starting model and consistent gyration radius (Figure 2C). On the other hand, the lowest free energy conformations in the presence of Ca^2+^ were characterized by low RMS deviations and slightly compacted architecture when compared to the starting model (Figure 2D). Similarly, free energy landscapes that were projected as a function of intra protein and protein–solvent contact were also constructed (Appendix A). The lowest free energy conformation of Ca^2+^ bound Rv0297PGRS showed a higher number of intra protein contacts than protein alone. Further variations in the secondary structure contents along the simulation timeline were analyzed. In Rv0297PGRS alone, a gradual decline in the β-content along with concurrent gain in coil content of the system was observed during the last 50 ns of simulations period. However, in the presence of Ca^2+^, the secondary structure contents of the system remained invariable with respect to the starting conformation. 

### 2.3. Calcium Binds PGRS Domain with Micromolar Affinity 

Steady-state fluorescence titrations and fluorescence-based thermal shift assays were carried out to investigate the binding of Ca^2+^ and its stabilization effect on the PGRS domain (Figure 3). In fluorescence titrations, an apparent dissociation constant of 158.01 ± 12.37 µM was observed for Ca^2+^ binding, with rRv0297PGRS indicating strong metal–protein association (Figure 3A). However, no such binding was observed with Mg^2+^ ions indicating the specificity of rRv0297PGRS for Ca^2+^ ions (Appendix A). With increasing Ca^2+^ concentrations during titrations, no concomitant red/blue shifts were observed. Consequently, we observed a slight stabilization of rRv0297PGRS upon initial binding with Ca^2+^. The melting temperature (T_m_) of Ca^2+^ bound rRv0297PGRS was ~2.5 °C higher than alone (45.7 ± 2.5 °C and 43.2 ± 0.5 °C, respectively), indicating a slight stabilization in the presence of Ca^2+^ ions (Figure 3B).

### 2.4. Ca^2+^ Bound PGRS also Stabilizes TLR4 Architecture

To investigate the potential role of Ca^2+^ in (TLR4)_2_ stabilization, previously simulated Rv0297PGRS in the presence and absence of Ca^2+^ ions was subsequently docked onto (TLR4)_2_ while using ClusPro (Appendix A). Interestingly, Rv0297PGRS pre-simulated in presence of docked Ca^2+^ ions showed a lower free energy score ΔG of −1339 as compared to ΔG of −1009 for pre-simulated Rv0297PGRS in the absence of Ca^2+^ ions (Figure 4A,B). This could be attributed to a decrease in the gyration radius of PGRS in Ca^2+^ bound state towards the end of the simulation period. The comparatively compact architecture of Ca^2+^ remodeled PGRS could snugly fit into (TLR4)_2_ binding pockets when compared to Ca^2+^ free PGRS. 

Further, the stability of (TLR4)_2_ in the presence of Ca^2+^ free and Ca^2+^ bound PGRS was ascertained using MD simulations. (TLR4)_2_ architecture stability was analyzed by constructing free energy landscapes that were projected onto its backbone RMSD and gyration radius, variations in distance between its C-terminal domains (CTD) and inter-CTD angle, and Cα fluctuations averaged over the whole simulation trajectory. In simulations of Ca^2+^ free Rv0297PGRS-(TLR4)_2_, FEL showed a single low energy basin corresponding to (TLR4)_2_ structure with gyration radius (Rg) of 4.25 nm and backbone RMSD of 0.4 nm from the starting model (Figure 2A). In the presence of Ca^2+^ ions, multiple low free energy metastable states of (TLR4)_2_ were observed (Figure 4B). These were characterized by similar backbone RMS deviation (~0.3–0.4 nm), albeit slightly low Rg (approximately 4.10–4.15 nm). Correspondingly, porcupine analysis (which depicts the magnitude and direction of atomic motions during simulations) showed minimal Cα fluctuations in (TLR4)_2_ in the presence of Ca^2+^ bound Rv0297PGRS, while these were substantially higher in Ca^2+^ free Rv0297PGRS, suggesting an overall stabilization effect conferred by metal-bound Rv0297PGRS. Additionally, it was observed that CTD were also brought into close apposition by Ca^2+^ bound PGRS. This was further confirmed by reduced inter-CTD distance (approximately 3.8 nm) and an acute inter-CTD domain angle (approximately 20°) of (TLR4)_2_ in the presence of Ca^2+^ bound PGRS (Figure 4C,D). However, in the absence of metal-bound PGRS, the average inter-CTD distance was substantially higher (approximately 4.2–4.4 nm) with a shallow inter-CTD angle of approximately 100°. Apposition of (TLR4)_2_ CTD and stabilization conferred by Ca^2+^ bound PGRS could be attributed to a larger surface interaction area between protein molecules. H-bond analysis over the simulation period showed a higher number of H-bonds per time frame for (TLR4)_2_-Rv0297PGRS system in the presence of Ca^2+^ ions than in their absence (Appendix A). 

### 2.5. Interaction of Rv0297PGRS with Macrophage TLR4 Is Stabilized in the Presence of Ca^2+^ Ions

In our previous report, we showed that Rv0297PGRS interacts with TLR4 of host macrophages to initiate the ER stress-mediated responses [21]. Because the Rv0297PGRS shows high binding affinity with calcium ions, in vitro immunofluorescence assays were performed to investigate interactions of rRv0297PGRS with TLR4 of macrophages in the presence and absence of calcium ions. Calcium free environment was achieved using 0.5 mM and 1 mM of EGTA during cell culture treatments. RAW264.7 macrophages were treated with 20 µg/mL of rRv0297PGRS protein for 2 h, either in the presence or absence of Ca^2+^ ions and subsequently immune-stained with anti-Rv0297PGRS sera followed by immunostaining with Alexafluor555 conjugated secondary antibody (Life Technologies, Waltham, MA, USA). Immunofluorescence images depicted the presence of red puncta over the surfaces of RAW264.7 cells when incubated with rRv0297PGRS protein for 2 h (Figure 5, Panel 2). The red puncta on cell surfaces could be an indication of interacting complexes on the cell surface.

In contrast, this red puncta was either reduced or absent in the presence of 0.5 mM or 1 mM EGTA, respectively (Figure 5, Panel 3 & 5), indicating the loss of rRv0297PGRS interaction with host cell surface receptors. EGTA chelates the calcium ions that are present in culturing medium, therefore the stabilization of interacting complex of Rv0297PGRS with cell surface receptors is hindered, leading to reduced puncta formation over RAW264.7 macrophage cell surfaces. In both cases, these interacting complexes were re-established by supplementing the culture medium with 2 mM CaCl_2_ (Figure 5, Panel 4 & 6). It indicates that the Rv0297PGRS proteins functions better in the presence of calcium ions. The interaction of Rv0297PGRS with TLR4 of the host cell surfaces had been observed in previous study [21]. These observations depict calcium-dependent interactions of rRv0297PGRS with the host cell receptor TLR4.

### 2.6. Downstream Effect of Rv0297PGRS Is Enhanced in the Presence of Calcium 

TLR4 agonists, such as Rv0297PGRS, lead to the production of NO from host cells via its interactions with TLR4 extracellular domains [21,47]. Based on immunofluorescence results, the calcium-dependent downstream production of NO from host macrophages in response to rRv0297PGRS protein treatment was also evaluated. It was observed that the rRv0297PGRS protein induces the production of NO species from treated macrophages in a concentration dependent manner. However, the production of NO was reduced post chelation of Ca^2+^ ions (Figure 6, light blue bars). In contrast, the replenishment of calcium leads to 20–30% enhanced production of NO from host macrophages when incubated with rRv0297PGRS protein as compared to the control (Figure 6, light green bars). Positive control thapsigargin induced NO production was not found to be dependent on calcium concentrations. These results suggest that the downstream effect of rRv0297PGRS via its interaction with TLR4 is enhanced in the presence of calcium ions. 

### 2.7. Rv0297PGRS Induces the Production of Pro-Inflammatory Cytokines in Calcium-Dependent Manner

We measured the TNF-α and IL-12 levels in activated THP-1 macrophages incubated with rRv0297PGRS protein either in the presence or absence of Ca^2+^ ions in order to investigate the likely role of calcium in the modulation of host immune responses through Rv0297PGRS. The production of TNF-α and IL-12 from THP-1 macrophages incubated with ranging concentrations of rRv0297PGRS protein was observed. However, this production of cytokines was found to be depleted in a calcium-free environment (Figure 7A,B, 3rd group; light green bars). This depicts that the Rv0297PGRS mediates the downstream effects in the presence of calcium only. Moreover, the production of TNF-α and IL-12 was regained upon replenishment of the culture medium with Ca^2+^ ions using 2mM CaCl_2_ (Figure 7A,B, 4th group; grey bars). We did not observe significant changes in the cytokine levels when the THP-1 cells were incubated with either 100nM LPS (positive control) or 20 μg/mL BSA (negative control). The levels of pro-inflammatory cytokines in rRv0297PGRS stimulated THP-1 cells were upregulated in a calcium-replenished medium as compared to the calcium depleted medium. The sole increment in calcium ions (without rRv0297PGRS protein) did not lead to any significant cytokine release (Figure 7A,B, dark blue and light blue bar in “0”). These results indicate the likely role of calcium in the immune responses that are generated through Rv0297PGRS.

## 3. Discussion

A multifaceted association between microbial components and host cellular responses will offer a varied degree of bacterial fitness and virulence [48]. 

The differential expression of few PE_PGRS proteins (PE_PGRS 3, 7 and 60) under various stresses, such as nutrient starvation, low pH, and low oxygen, have been reported [46]. PE_PGRS proteins, implicated in antigenic variation and immune evasion [5,7,49,50], have been observed to be secreted via the ESX-5 secretion system [51,52]. A recent finding described the importance of the PGRS domain of PE_PGRS5 in conferring enhanced survivability to non-pathogenic recombinant *M.smegmatis* in hypoxic and acidic conditions and increased bacillary load in infected host macrophages [45]. PE_PGRS proteins have also been implicated in modulating alveolar macrophage functioning by altered pro- and anti-inflammatory responses during *M.tb* infections. The downregulation of IL-6 and IL-1β along with up regulation of IL-12p40, and IL-10 has been evident in response to PE_PGRS18/62 [53,54]. PE_PGRS41 expressing *M.smegmatis* causes decreased expression of TNF-α, IL-6, and IL-1β cytokines, thereby altering macrophage functions [55]. Another recent report suggested the importance of PE_PGRS31 in promoting mycobacterial survival within host macrophages by the inhibition of TNF-α secretion via downregulating NF-κB- signaling cascade [56]. Interestingly, PE_PGRS31 has also been found to intervene in host lipid metabolism by disrupting arachidonic acid pathway via its interacting with S100A9 macrophage protein [56]. These findings suggest unambiguous roles of specific domains of PE_PGRS proteins, which might result in differential outcomes of host pathogen interaction. 

Exploration of the role of PGRS domain of PE_PGRS protein domains may aid in a better understanding at the host-pathogen interface. The PGRS domain of Rv0297 (a PE_PGRS protein, Rv0297PGRS) coded by *M.tb* genome, has been investigated in this study for its calcium-dependent action on TLR4 receptors of host macrophages. A total of 56 out of the 61 PE_PGRS proteins of *M.tb* proteome were found to consist of repetitive nonapeptide calcium-binding motifs of GGXGXD/NXUX type, which can form parallel β-roll or parallel β-sheets (46). The GGXGXD motifs were also observed in other Ca^2+^ binding proteins, e.g., RTX toxin and alkaline protease that are secreted by Gram-negative *Magnetococcus sp. MC-1* and *Pseudomonas aeruginosa* [41,42,57], where calcium-binding could induce structural order in these intrinsically disordered proteins, aiding in protein secretion and virulence (42,43). Pathogens, like *M.tb,* use intrinsically disordered proteins (8), notably the transition from unstructured to structured secondary structures, in hijacking and perturbing host cellular responses for the progression of disease pathogenesis [35,58,59,60,61]. However, the number and type of nine amino acid motifs can vary from protein to protein [46]. For example *M.tb* Rv1818c contains 10 motifs, Rv3653 has a single binding motif [62], while the Rv0297PGRS harbored 15 nonapeptide calcium-binding motifs (Appendix A). PE_PGRS33 and PE_PGRS61 have both been demonstrated to initiate the production of IL-10 via their interaction with TLR2 in a Ca^2+^ dependent manner [62]. The focal point of interactions of *M.tb* with host cell receptors may depend on extracellular Ca^2+^ ions that, in turn, are associated with downstream signaling events [63]. In light of these facts, it was of key interest to study the Ca^2+^ binding mycobacterial proteins that may be involved in host–pathogen interactions. Deciphering the role of the PE_PGRS family of proteins in lieu of possible Ca^2+^ binders may reveal new aspects of the pathophysiology of TB.

The initial step in the host–pathogen interaction involves the association between PAMPs (pathogen-associated molecular patterns) from the microbes and PRRs (pattern recognition receptors) from host cells [64]. This interaction may be enhanced in the presence of different stabilizers and co-factors. TLRs serve as the first ligands for sensing the microbial components, which, in turn, is an essential step for modulating the host responses [64,65,66]. *M.tb* secretes several proteins that interact with TLRs of host macrophages and modulate the cellular cascades, such as the production of ROS or NO intermediates [67], host cell apoptosis [68], antigen presentation [69], and phagosomal acidification [70] for initiating disease pathogenesis. Virulence and pathogenesis of *M.tb* involve TLRs, such as TLR1, TLR2, TLR4, and TLR9, and their associated signaling pathways [71]. Mycobacterial cell wall and secreted components have also been indicated to induce TLR2 and TLR4 dependent responses [72]. TLR4 mediated signaling aids in sustaining the balance of necrotic and apoptotic macrophages [37]. We earlier demonstrated the localization of Rv0297PGRS to ER of host cells, thereby evoking ER stress-mediated response. The stress response was dependent on the interactions between Rv0297PGRS and TLR4 [21]. Rv0297PGRS was also shown to affect calcium homeostasis and hinders phagolysosomal maturation [21]. Through in-silico and in-vitro correlations, our study revealed calcium-dependent stabilization of the Rv0297PGRS and its effect on TLR4 responses. 

Ca^2+^ is a major secondary messenger and it is an important co-factor that is involved in host responses [63]. MD simulations and fluorescence assays revealed the stabilization effect of Ca^2+^ ions on Rv0297PGRS (Figure 2 and Figure 3). Further, Rv0297PGRS stabilized in the presence of Ca^2+^ ions could act as a TLR4 agonist. It was recently shown that (TLR4)_2_ agonists aid in downstream signaling by reducing atomic fluctuations and stability at its C-terminal end, thereby promoting the association of positive effector molecules to this domain [73]. In our analysis, we observed a similar reduction in Cα fluctuations at the C-terminal end of (TLR4)_2_, only in the presence of Rv0297PGRS bound to Ca^2+^ ions, thus reinforcing its Ca^2+^ dependent modulation of TLR4 signaling. In-vitro immunofluorescence studies using rRv0297PGRS further confirmed the calcium dependent interaction with TLR4 receptors (Figure 5). Upon chelation of Ca^2+^ using EGTA, the rRv0297PGRS-TLR4 interaction signal was perpetually lost in host macrophages. Replenishing the medium with CaCl_2_ restored the Rv0297PGRS-TLR4 interaction. 

Rv0297PGRS also induces NO production from macrophages in a TLR4 dependent manner [21]. Our study was extended to examine the role of calcium ions in activating TLR4 in order to evoke downstream responses, such as NO production. It was observed that, upon chelating calcium from the extracellular environment, the NO production by macrophages in response to rRv0297PGRS protein treatment was reduced (Figure 6). In comparison to this, the NO levels were upregulated from macrophages when Rv0297 was pre-incubated with CaCl_2_ that increased the efficiency of Rv0297PGRS to activate host cell receptors (Figure 6). 

Our results demonstrated that the functions employed by *M.tb* through the PGRS domain of Rv0297 are calcium-dependent. Rv0297PGRS is able to bind with calcium ions, as evident by MD simulations and intrinsic fluorescence spectroscopy. In-silico MD simulations also described the stabilizing effect of calcium on Rv0297PGRS. The Ca^2+^ docked Rv0297PGRS protein was able to efficiently provide a stabilizing effect to the C-terminal domain of TLR4 and, thus. allows it to become activated. TLR4, in turn, gets activated through the Rv0297PGRS protein in the presence of calcium ions and it evokes a signaling process that leads to the generation of NO. NO production was decreased when calcium was depleted from the extracellular environment. Moreover, rRv0297PGRS evokes the release of TNF-α and IL-12 from host macrophages in calcium-supported manner.

In conclusion, Rv0297 encoded PE_PGRS5 has been found to be implicated in calcium-dependent host responses that are mediated via TLR4 during *M.tb* infection, which may be utilized for a better understanding of PGRS domains of PE_PGRS proteins (Figure 8). The calcium dependence property can be targeted for improvising therapeutics and vaccine strategies. 

## 4. Material and Methods

### 4.1. Prediction of Calcium-Binding Motifs in M.tb PE_PGRS Proteins 

Amino acid sequences of mycobacterial PE_PGRS proteins were retrieved from the UniprotKB database and the presence of Ca^2+^ binding motifs (GGXGXD/NXUX), where X is any amino acid and U is unipolar hydrophobic residue) was examined using regular expression in sequence analysis suite [74]. A comparative sequence alignment of PGRS domains from PE_PGRS5 and other PE_PGRS proteins (PE_PGRS33, PE_PGRS20 and PE_PGRS45) was carried out using Clustal Omega (https://www.ebi.ac.uk/Tools/msa/clustalo/ accessed on 24 March 2021). For the calculation of the surface electrostatic potential of Rv0297PGRS, the APBS electrostatics plugin of PyMol was used [75].

### 4.2. Prediction of Ca^2+^ Binding Sites and Structural Modeling of Rv0297PGRS-TLR4 Complex 

IonCom was also used to perform composite metal binding-site prediction employing both ab initio training and template-based transferals [76]. *M.tb* Rv0297 was subjected to comparative modeling using I-TASSER [77]. Z-scores (from i-tasser modelling) were used to check the correct domain folds and overall tertiary structure. Before subjecting the model to protein-protein docking and MD simulations, Ramachandran and ProSA analysis were performed to ensure correct starting configuration [78]. Ca^2+^ binding site predictions were also made using COFACTOR and COACH analysis during I-TASSER modeling. Further, the modeling of (TLR4)_2_-(Rv0297PGRS)_2_ hetero-tetramer was done using the ClusPro 2.0 protein–protein docking engine where TLR4 dimer [(TLR4)_2_, PDB id: 3vq2] and Rv0297PGRS were used as receptor and ligand, respectively [79]. In ClusPro docking, the Rv0297PGRS structure was extracted from the simulation trajectory and subsequently docked.

### 4.3. Molecular Dynamics Systems and Simulations 

Initially, to study the effect of calcium-binding to Rv0297PGRS, molecular dynamics (MD) simulations of Rv0297PGRS alone and in the presence of Ca^2+^, which corresponded to experimental setups, were carried out. In another set, to study the effect of Ca^2+^-Rv0297PGRS on TLR4 architecture, MD simulations of (TLR4)_2_-(Rv0297PGRS)_2_ were set up, both in the absence and presence of Ca^2+^. The Rv0297PGRS exclusive systems were run for 200 ns, while the (TLR4)_2_-(Rv0297PGRS)_2_ systems were run for 50 ns.

All of the simulations were performed using GROMACS 5.1 and GROMOS 54a7 all-atom force field [80,81]. Using periodic boundary conditions, the starting models were solvated in a periodic box with a SPC/E water model and 10 Å spacing from each edge of protein. Counter ions (Na^+^ or Ca^2+^) were added to neutralize and maintain ionic concentrations that were equivalent to that used in experimental setup. Subsequently, these were energy minimized using a steepest descent protocol, followed by an equilibration run of 1 ns for both NVT and NPT ensemble with positional restraints on the protein atoms. The systems were simulated at 310 K maintained by a Berendsen thermostat and pressure coupling employing a Parrinello-Rahman barostat using a one bar reference pressure and compressibility factor of 4.5e^−5^ bar, using isotropic scaling scheme. Electrostatic interactions were calculated using the Particle Mesh Ewald (PME) summation with 2 fs time step for each run. The resultant trajectories were analyzed using standard GROMACS tools. A collection of low amplitude dynamics from trajectories to carry principal component analysis of (TLR4)_2_-(Rv0297PGRS)_2_ systems was done by the diagonalization of the mass-weighted covariance matrix for the C-alpha atoms of (TLR4)_2_. The resultant trajectories were projected onto first eigenvector and the fluctuations were calculated between extreme projections to construct porcupine plots. Images were generated using PyMol.

### 4.4. Binding of Ca^2+^ with Rv0297PGRS

In order to generate recombinant Rv0297PGRS (rRv0297PGRS) protein, the Rv0297 gene was cloned in pET28a expression vector and purified, as described earlier [21]. The binding affinity (K_d_) of Ca^2+^ with rRv0297PGRS was calculated using fluorescence spectroscopy. All of the titrations were carried out in 20 mM Tris, 150 mM NaCl, pH 8.0 at 298 K using Perkin Elmer LS55 spectrophotometer. Fluorometric titrations were carried at constant rRv0297PGRS concentration (2μM) and increasing concentrations of Ca^2+^ up to 500 µM with a pre-scan delay of 300 s. The samples were excited at 290 nm and emission spectra were recorded from 290 nm to 400 nm. In order to determine K_d_, the fluorometric titration data were fitted with single site-specific binding module of GraphPad Prism v 6.0 using the following equation:Y = B_max_ × X/(K_d_ + X)(1)
where B_max_ is the maximum specific binding and X represents the Ca^2+^ concentration at each titration.

### 4.5. Assessment of Rv0297PGRS Stabilization/Destabilization by Ca^2+^

The intrinsic fluorescence-based thermal shift (FTS) assay was performed by monitoring changes in Trp/Tyr emission at 350 nm over the temperature range from 25 to 70 °C. The temperature increment step size was kept at 2 °C with a pre-measurement delay of 60 s. Excitation and emission slits were kept at 10/10. Thermal denaturation of rRv0297PGRS alone was done at the concentration of 2 μM and for Ca^2+^ bound state, 2 μM of protein was pre-incubated with 200 μM of Ca^2+^ for 60 min. prior to assay.

### 4.6. Secondary Structure Estimation of Rv0297PGRS 

Cloning and purification of rRv0297PGRS was carried out as described earlier [45]. Briefly, it was performed by on-column refolding of the Rv0297PGRS using Ni-NTA chromatography in phosphate-buffered saline, pH 7.4. The secondary structure was estimated using circular dichroism spectroscopy (CD) in a Jasco J-815 spectropolarimeter (Jasco Inc., Mary’s Ct, Easton, MD, USA). For FarUV CD (190–250 nm) measurements 10 µM rRv0297PGRS protein was placed in a quartz cell of 1 mm optical path length at 25 °C. Baseline correction was done using buffer alone with scan speed of 50 nm/min. All of the solutions were 0.22µm filtered prior spectral measurements. Secondary structure content from CD spectra was estimated using BeStSel (Beta Structure Selection) module [82,83].

### 4.7. Cell Culture 

The macrophage cell lines, RAW264.7 and THP-1, were maintained in Dulbecco’s modified Eagle’s medium (Invitrogen) and Roswell Park Memorial Institute RPMI 1640, respectively supplemented with 10% FBS (Invitrogen), penicillin (100 IU/mL), and streptomycin (100 μg/mL). Cells (1 × 10^5^/well) were seeded in 24-well plates and treated with different concentrations of rRv0297PGRS protein.

### 4.8. Immunofluorescence Staining

RAW264.7 cells were seeded and allowed to adhere on coverslips in 24-well plates, followed by treatment with 20 μg/mL rRv0297PGRS with or without ethylene glycol-bis (β-aminoethyl ether)-N, N, N′, N′-tetra acetic acid (EGTA), and calcium chloride (CaCl_2_). After 2 hrs. of incubation, the cells were fixed with 3.7% paraformaldehyde for 15–20 min. followed by blocking for 10 min. with 0.1% TritonX-100/1% BSA/PBS. The cells were labeled with respective primary and Alexaflour-conjugated secondary antibodies [84]. Immunofluorescence imaging was done using an Olympus Fluoview Laser-scanning microscope. 

### 4.9. Nitrite Quantitation in Macrophages

RAW264.7 cells were treated with purified rRv0297PGRS protein for 30 h. Calcium ions chelation in culture medium was achieved by using EGTA (specific Ca^2+^ chelator). After the required period of incubation, cell-free supernatant (150 µL) was mixed with 50 µL of Griess reagent for 30 min. The nitrite concentration was measured using sodium nitrite as a standard. Plates were read at 540 nm.

### 4.10. Cytokine Assessment in Macrophages

The RAW264.7 cells were treated with rRv0297PGRS protein (0–10 µg/mL) for 30 h either in the presence or absence of Ca^2+^ ions. Cell-free supernatant was collected and TNF-α and IL-12 concentrations were measured using ELISA Kit (eBiosciences), as per the manufacturer’s instructions. Plates were read at 450 nm.

### 4.11. Statistical Analysis

All of the data are expressed in the form of mean ± standard deviation (S.D.) derived from three different groups of independent experiments, using GraphPad Prism 6.0 software. A one-way analysis of variance (ANOVA) was performed, followed by Dunnett’s post hoc test, in order to calculate the statistical significance at *p*-value < 0.05.

## Figures and Tables

**Figure 1 ijms-22-09390-f001:**
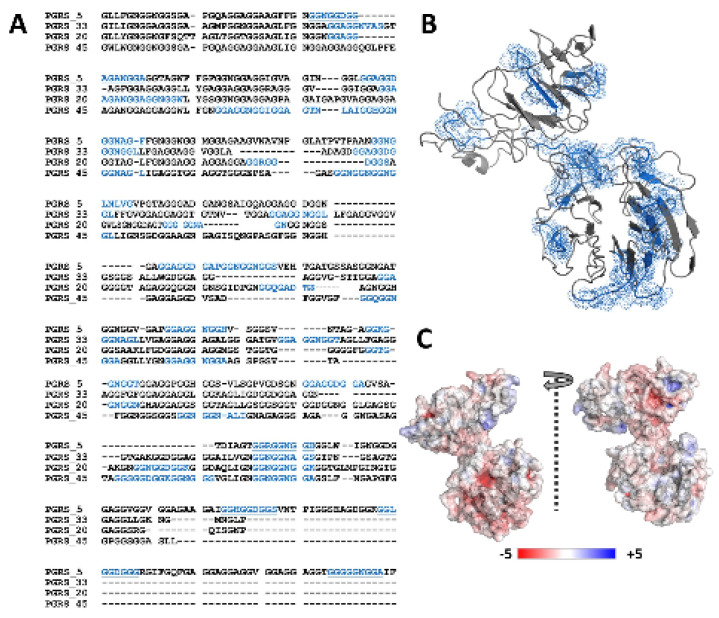
Sequence and structural features of Rv0297PGRS. (**A**) Sequence alignment of PGRS domains from various PE_PGRS proteins of *M.tb*. PGRS_5 (Rv0297PGRS) is aligned with a known calcium-binding protein PGRS_33 (Rv1818cPGRS) and other PGRS proteins containing putative calcium-binding motifs; PGRS_20 (Rv1068c) & PGRS_45 (Rv2615cPGRS) of *M.tb.* Highlighted blue are putative calcium-binding regions. (**B**) Modeled structure of Rv0297PGRS monomer showing spatial distribution of calcium-binding motifs (Blue). (**C**) Surface electrostatic potential of Rv0297PGRS calculated by APBS. Positively charged regions are colored in blue and negatively charged regions in red.

**Figure 2 ijms-22-09390-f002:**
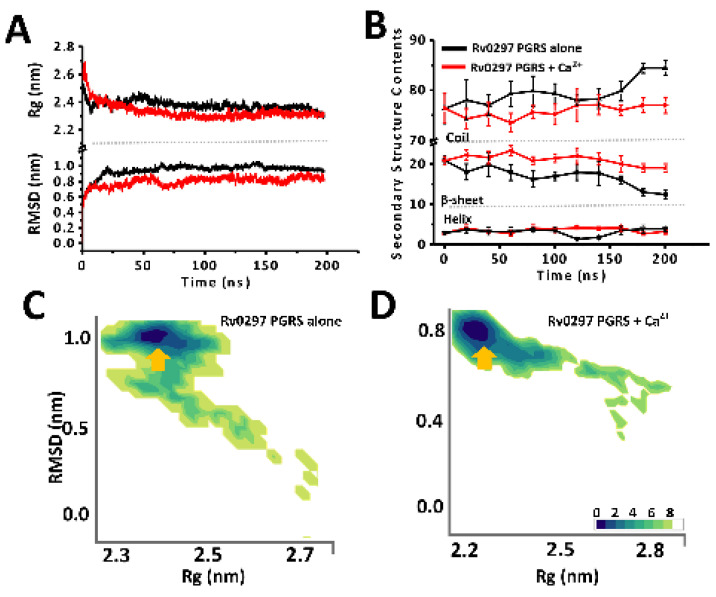
Molecular dynamics (MD) simulation outcomes of Rv0297PGRS alone and in the presence of calcium ions. (**A**) Variations in backbone RMSD and Cα gyration radius along the 200 ns trajectory and (**B**) Transitions in secondary structure elements of both systems, obtained after simulations of Rv0297PGRS alone (Black) and in presence of Ca^2+^ ions (**C**,**D**) Free energy landscapes (in kJ/mol) projected onto backbone RMSD and gyration radius (Rg) averaged over the whole simulation trajectory, for Rv0297PGRS alone and in presence of Ca^2+^ ions, respectively. Arrows indicate low free energy basins corresponding to metastable stables thata are achieved by Rv0297PGRS in both simulations.

**Figure 3 ijms-22-09390-f003:**
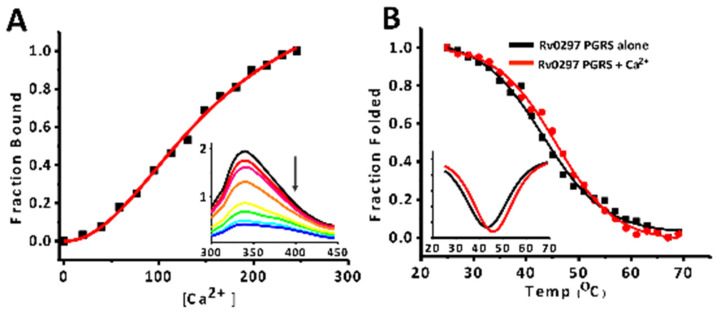
**Fluorescence titrations and thermal shift assay.** (**A**) Nonlinear curve fit of fluorescence emission maximum to determine Kd for Rv0297PGRS with Ca^2+.^ Inset shows fluorescence emission spectra (300–450 nm) of each titration of protein with increasing concentration of Ca^2+^. (**B**) Non-linear curve fitting analysis of fluorescence-based thermal shift assays to determine melting temperature (Tm) of Rv0297PGRS alone (Black) and in presence of Ca^2+^ (Red). Insets show first derivative of respective Tm curves.

**Figure 4 ijms-22-09390-f004:**
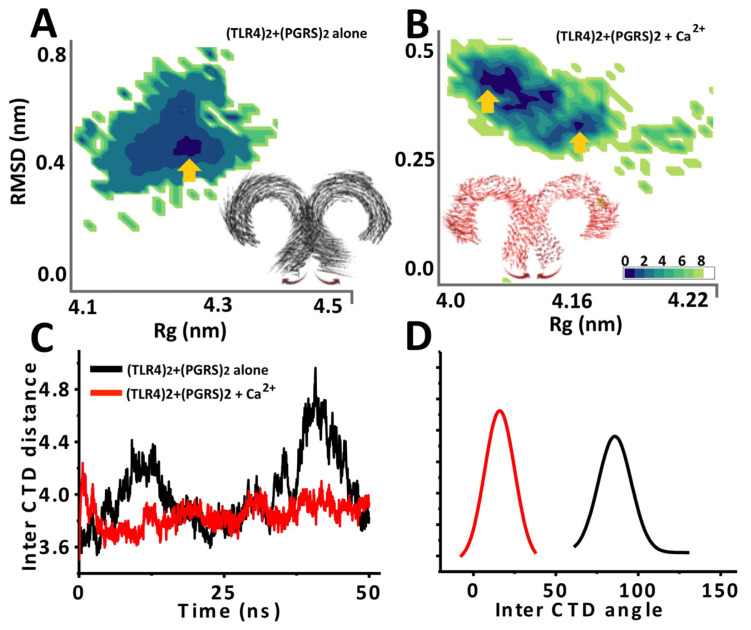
MD simulation outcomes of (TLR4)_2_-(Rv0297PGRS)_2_ alone and in the presence of calcium ions. (**A**,**B**) Free energy landscapes (KJ/mol) projected as a function of two macroscopic principal components Rg and Backbone RMSD for (TLR4)2 Alone and (TLR4)_2_-(Rv0297PGRS)_2_, respectively. Arrows indicate low free energy basins corresponding to metastable stables achieved by (TLR4)2 in both sets of simulations. Insets show porcupine plots deduced from MD simulations of (TLR4)_2_ complexed with Rv0297PGRS both in the absence and presence of Ca^2+^ ions. (**C**) Variations in inter C-terminal domain distance and (**D**) Inter C-terminal domain angle between two TLR4 monomers, alone and in presence of Ca^2+^ ions, averaged over the last 10 ns of the trajectory.

**Figure 5 ijms-22-09390-f005:**
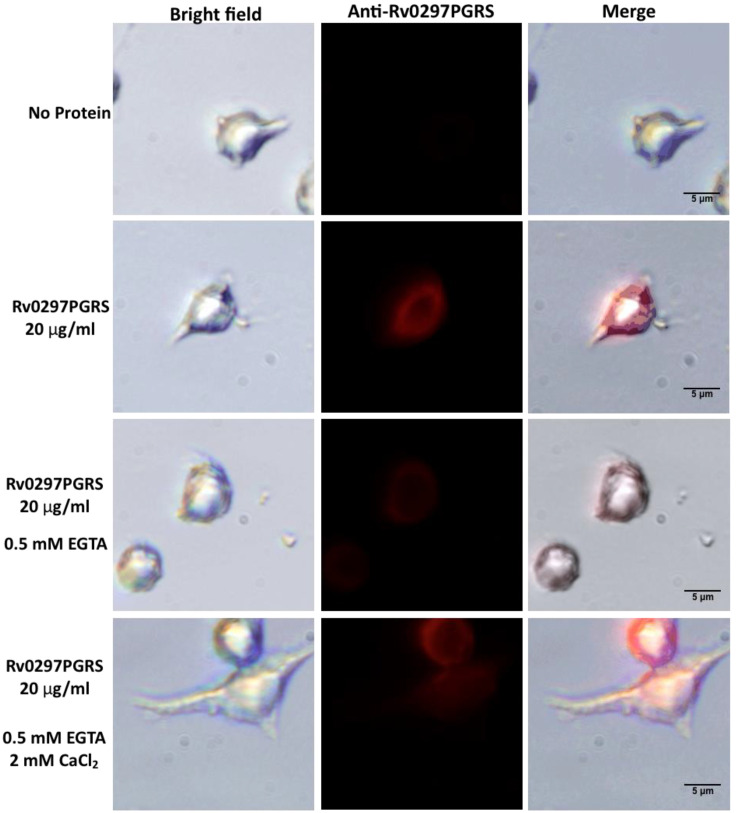
In vitro binding of Rv0297 PGRS with TLR4 of macrophages in the presence of Ca^2+^ ions. RAW 264.7 cells incubated with recombinant Rv0297PGRS protein in complete DMEM (consisting Ca^2+^ ions) for 2 h and stained with anti-Rv0297PGRS sera (Panel 2). Untreated cells were used as negative control (Panel 1). Calcium was removed from medium using 0.5 mM and 1 mM EGTA (third and fifth panel, respectively), followed by incubation with rRv0297PGRS protein and staining with anti-Rv0297PGRS sera. Ca^2+^ levels were replenished in the culture medium using 2mM CaCl_2_. Alexa Fluor 555 fluorescent images were captured for TLR4 and Rv0297PGRS interaction and merged images were created (right column). Red colored puncta formation over RAW264.7 cell surfaces depicts the interaction of Rv0297PGRS with TLR4 of macrophages (2nd, 4th and 6th panel). The puncta formation (or interacting complexes) was reduced in case of using EGTA (3rd and 5th panel). Magnification, X 600.

**Figure 6 ijms-22-09390-f006:**
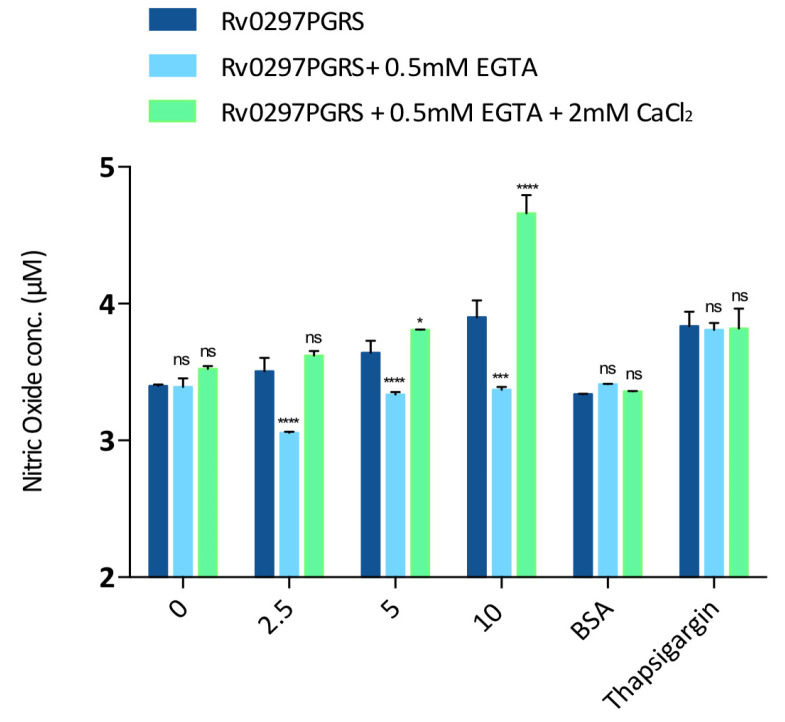
Rv0297 PGRS enhances the production of NO from macrophages in the presence of calcium ions. Assessment of NO production from Rv0297 PGRS stimulated RAW264.7 cells in the presence and absence and absence of Ca^2+^ ions was performed. Calcium’s presence was observed to enhance Rv0297 PGRS induced NO secretion from treated macrophages. In comparison to that, the removal of Ca^2+^ leads to reduced production of NO in response to rRv0297PGRS protein. One micromolar thapsigargin was used as positive control. BSA (10 μg/mL) was used as negative control. Data were plotted in NO concentration. All values were represented as means ± SDs from three independent experiments. *p* values for *, ***, ****, and ns are <0.05, <0.01, <0.001, and >0.05, respectively.

**Figure 7 ijms-22-09390-f007:**
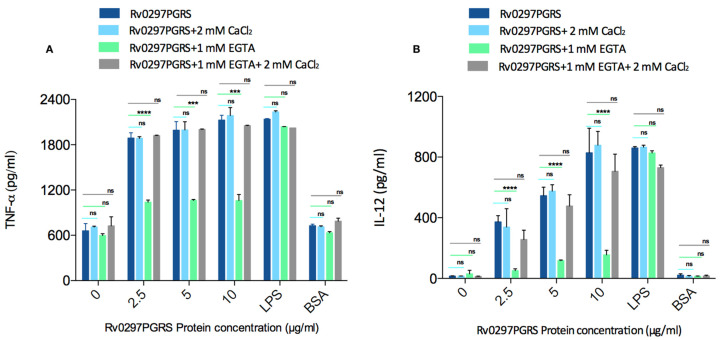
Rv0297 PGRS induces cytokine production in calcium-dependent manner. Calcium supported rRv0297 PGRS induced TNF-α (**A**) and IL-12 (**B**) production in THP-1 macrophages. Normal cell culture medium (DMEM) consists of calcium chloride salt. Calcium free environment was achieved using 1 mM EGTA in the culture medium. Replenishment of Ca^2+^ ions was done by using 2mM CaCl_2_ along with 1mM EGTA in the experimental medium to overcome the effect of EGTA. A range of Rv0297PGRS protein was used to stimulate THP-1 macrophages under different types of experimental conditions mentioned above. Data were plotted as cytokine concentration in pg/mL. All of the values were represented as means ± SDs from three independent experiments. *p* values for ***, ****, and ns are <0.01, <0.001, and >0.05, respectively.

**Figure 8 ijms-22-09390-f008:**
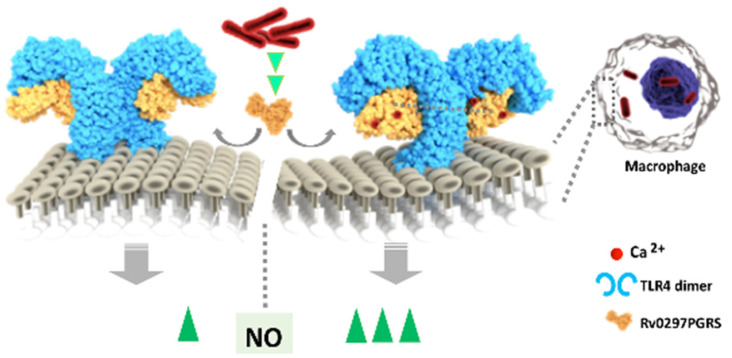
Schematic representation of Calcium dependent functions of Rv0297PGRS. The binding of Ca^2+^ with Rv0297PGRS stabilizes the protein and enhances its interaction with TLR4 of macrophages. The Ca^2+^ stabilized binding of Rv0297PGRS with the macrophage surface receptor of macrophages increases the production of NO and pro-inflammatory cytokines. These calcium binding motifs in the PGRS domain may aid in the stabilization of the unstructured/disordered PE_PGRS proteins, involved in virulence and pathogenesis of tuberculosis.

## Data Availability

Not applicable.

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
