# Peer review of "PGRS Domain of Rv0297 of Mycobacterium tuberculosis Functions in A Calcium Dependent Manner"

_ijms, 2021, doi:10.3390/ijms22179390_

Round 1

Reviewer 1 Report

In this research article, authors used molecular dynamics simulations and fluorescent microscopy to demonstrate that the Rv0297PGRS protein of mycobacterium tuberculosis is stabilized by Ca2+. Authors concluded that this stabilization enhances the interaction between the Rv protein and the TLR4 receptors localized on macrophage surface. This interaction plays a significant role to produce NO and release TNFa and IL-12. The research could be improved and some control seems missed. The manuscript is not suitable for publication in this current form.

Major points:

  • Results presented in figure 5 can be improved. Authors should use confocal microscopy and develop tools to study colocalization between TLR4 and Rv0297 proteins. It could be interesting to detect TLR4 at the membrane. Are we sure that the protein binds only TLR4 at the surface? Could you use TLR4 deficient macrophages (or siRNA) as negative control? Do we know how TLR4 is express by RAW cells? Moreover, do you have the opportunity to quantify (staining intensity per cells) the interaction between Rv0297 at the surface depending on Ca2+ concentration (kind of dose-response)?
  • Results presented in figure 6: Authors should use TLR4 deficient macrophages as negative control. Moreover, authors mentioned that TLR4 stimulation could induce cell death. Do we know if the tested conditions could induce apoptosis leading to a variation in NO production?
  • Results presented in figure 7: authors mentioned in the discussion that PGRS are implicated in other cytokines production. What is the impact of Rv0297 on the production of IL-6 and IL-1beta in macrophages.
  • General comment: Authors must justify why results presented in fig 5 and 6 were down on RAW cells when results presented in fig 7 were down on THP1 cells. Are you able to conduct all experiments on both cell types?

Author Response

We thank reviewer for the insightful comments. 

A point to point response along with revised manuscript are attached. 

Reviewer 2 Report

The article "PGRS domain of Rv0297 of Mycobacterium tuberculosis functions in a calcium-dependent manner" describes
the function of PGRS domain via bioinformatics and experimental analysis.
The authors have performed simple and nice experiments to show that PE_PGRS protein depends on Ca2+ for their function.
I made specific comments in the PDF document and those need to be addressed to improve the presentation of the results.
In brief:
The abstract can be presented in a better way instead of starting with "our results showed..
The introduction needs careful checking for recent reference of e.g. WHO 2020 report for minor English errors and abbreviations.
The methods are adequate but the results section is difficult to read and could be presented in a better way (specific comments are in the file).
The quality and presentation and details of the figures are very poor.
The discussion sounds too detailed not related to the results especially in the beginning and can be presented in a precise manner and focussed.
Figure 8, hypothesis? can be presented maybe elsewhere? is it a conclusion? not sure.

Author Response

(The authors gave the same response as above.)

Reviewer 3 Report

PGRS domain of Rv0297 of Mycobacterium tuberculosis functions in calcium dependent manner

Dear author and editor:

The author talked about the role of calcium ion in enhancing the interaction of PGRS domain of Rv0297 of Mycobacterium tuberculosis with TLR4. this article could be published after a revision.

I have some comments on it:

  • In some sentences inside the text, the author used different font style like in the line 160 and repeat it many time with other line please check. it should be all with the same style. 
  •  What are the molecular weight of the RV0297? do you have an immunoblot?
  • Did you free the purified recombinant protein from LPS?
  • Did you confirm that more calcium binding motifs present in the PGRS domain the more affinity to calcium ion or do you have any related data? 
  • PE_PGRS33 contains 10 motifs you didnot write how many motifs in the PE_PGRS45 and PE_PGRS20?
  • You can discuss in the part of the discussion, why Mtb conserve this motifs? i wonder if this motif presents in other MTB complex strains like M. bovis and if it is essential for M. tuberculosis pathogenesis?
  • The image in the figure (1) is not clear. 
  • did you try to use another PE_PGRS protein as a control to evaluate the stability increase obtained by calcium ion?
  • Why did you choose the magnesium as another positive control.
  • In the figure 3 the stabilization increase is significant or not? you should indicate.
  • Does Rv0297 have a unique role during Mtb infection?
  • The effect of protein stabilization is specific for TLR4 or affect also other TLRs.
  • Do you think that the effect of calcium ion to stabilize TLR4PGRS interaction as a results of an electrostatic interaction between positive ion (calcium) and the negative overall charge of the cell surface? did you try other protein as a negative control.
  • Why you did not try to verify the role of calcium in RvPGRS-TLR4  macrophages interaction during Mtb infection or infection of macrophages with M. smegmatis expressing Rv0297?
  • In the figure (5). why you did not use other condition like (without EGTA, without EGTA but with calcium)?
  • In the studying of the effect of calcium on pro-inflammatory cytokines, why you did not study the effect of the calcium alone then you can see the effect of adding more calcium, because is not clear if the EGTA have a second role prevent binding of protein with other ion even if with lower affinity like magnesium, so when you have added EGTA probably suppressed the effect of EGTA and launched the effect of the PGRS domain.
  • The author said that the RV0297 induces cytokines production in a calcium dependent manner. this effect is significant or not? please indicate it on the figure and on the figure legends. 
  • Why you did not use another positive ion as a negative control? 
  • Do you have any information about the localization of the protein RV0297? it is on the surface or it is a secreted protein?
  • In your previous work, you find the intracellular effect of this protein.  i can suggest that this protein have an extracellular role as well?

Thank you very much, best regards

Author Response

(The authors gave the same response as above.)

Round 2

Reviewer 1 Report

The authors answered all my concerns.